# Implications of Crosstalk between Exosome-Mediated Ferroptosis and Diseases for Pathogenesis and Treatment

**DOI:** 10.3390/cells12020311

**Published:** 2023-01-13

**Authors:** Zixuan Zhou, Benshuai You, Cheng Ji, Leilei Zhang, Feng Wu, Hui Qian

**Affiliations:** 1Jiangsu Key Laboratory of Medical Science and Laboratory Medicine, Department of Laboratory Medicine, School of Medicine, Jiangsu University, Zhenjiang 212013, China; 2Zhenjiang Key Laboratory of High Technology Research on Exosomes Foundation and Transformation Application, Zhenjiang 212013, China

**Keywords:** exosomes, ferroptosis, ferroptosis therapy

## Abstract

Ferroptosis is a type of iron-dependent cell death caused by ferrous iron overload, reactive oxygen species generation through the Fenton reaction, and lipid peroxidation, leading to antioxidative system dysfunction and, ultimately, cell membrane damage. The functional role of ferroptosis in human physiology and pathology is considered a cause or consequence of diseases. Circulating exosomes mediate intercellular communication and organ crosstalk. They not only transport functional proteins and nucleic acids derived from parental cells but also serve as vehicles for the targeted delivery of exogenous cargo. Exosomes regulate ferroptosis by delivering the biological material to the recipient cell, affecting ferroptosis-related proteins, or transporting ferritin-bound iron out of the cell. This review discusses pathogenesis mediated by endogenous exosomes and the therapeutic potential of exogenous exosomes for ferroptosis-related diseases. In addition, this review explores the role of exosome-mediated ferroptosis in ferroptosis-related diseases with an emphasis on strategies for engineering exosomes for ferroptosis therapy.

## 1. Introduction

Ferroptosis is a newly discovered regulated form of cell death. Its morphological, biochemical, and genetic characteristics differ from those of other types of cell death [1]. Morphologically, ferroptosis is characterized by the presence of intact nuclei, smaller mitochondria with decreased cristae, and ruptured mitochondrial membranes [2]. Iron metabolism, the formation of polyunsaturated fatty acids phospholipids (PUFA-PLs), and lipid peroxidation are the main factors that drive ferroptosis. The production of reactive oxygen species (ROS) by excessive iron through the Fenton reaction is crucial for the initiation and execution of ferroptosis. By contrast, increased iron storage and decreased iron intake may prevent iron-dependent lipid ROS accumulation. Three cellular defense mechanisms with different subcellular localizations play substantial roles in preventing ferroptosis, namely the glutathione peroxidase 4 (GPX4)-glutathione (GSH) system in the cytoplasm and mitochondria, the ferroptosis suppressor protein 1(FSP1)–CoQH2 system in the plasma membrane, and the dihydroorotate dehydrogenase (DHODH)–CoQH2 system in the mitochondria. The GPX4–GSH and FSP1–CoQH2 pathways work in parallel to neutralize lipid peroxides [3], whereas the DHODH–CoQH2 pathway is a newly discovered glutathione-independent defense system for ferroptosis in mitochondria. DHODH, located in the inner mitochondrial membrane, inhibits ferroptosis in the mitochondria by producing CoQH2 [4]. CoQH2 acts as a radical-trapping antioxidant to prevent lipid peroxidation and thus inhibits ferroptosis. The analyses of the aforementioned three mechanisms are integral to the study of ferroptosis.

Ferroptosis affects the development of various diseases. For example, suppression of ferroptosis by iron chelators or lipophilic radical-trapping antioxidants promotes the repair of damaged tissues, especially in in vivo models of ischemia/reperfusion (I/R) injury [5,6]. The induction of ferroptosis by depleting glutathione, inhibition of GPX4, and disruption of iron homeostasis improved the susceptibility of tumor cells to chemotherapy [7,8]. Therefore, targeting the development of diseases by regulating the ferroptosis of cells has attracted considerable attention [9], and developing therapies based on ferroptosis, hereinafter called as ferroptosis therapy (FT), is vital.

In ferroptosis-related diseases, exosomes regulate ferroptosis. Exosomes are membranous vesicles with a diameter of approximately 40 to 160 nm and are secreted by various cells and body fluids [10]. Exosomes that are present in serum, tumor, and immune cells are usually involved in the pathogenesis of many human diseases [11,12,13,14]. The low toxicity and biodegradability of stem cell-derived exosomes make them reliable candidates for the treatment of several diseases [11,12,13]. Our laboratory was the first to report the potential role of human umbilical cord MSCs (hucMSCs) in the repair of liver fibrosis [14]. The utility of exosomes in diagnosis and therapy lies primarily in their delivery of molecular entities [15,16]. Exosomes are involved in regulating several cell death pathways, such as autophagy, apoptosis, pyroptosis, and ferroptosis [9,17,18,19]. For example, exosomes can be used as drug delivery carriers to improve chemoresistance by regulating apoptosis signaling pathways [20]. Moreover, exosomes can release exogenous cargo into target cells to regulate the autophagic level of target cells [21]. During the transport of microRNAs to damaged cells, exosomes from mesenchymal stem cells regulate pyroptosis [22]. Exosome-derived noncoding RNAs (ncRNAs) regulate ferroptosis-related proteins [23,24]. Ferritin-containing multivesicular bodies (MVBs) or exosomes are speculated to excrete iron from cells and thus prevent the onset of ferroptosis. Genetic engineering or the chemical modification of the surface of exosomes may improve their targeting efficiency and specificity for cargo delivery in FT. Mechanisms underlying exosome-mediated ferroptosis should be elucidated, and effective therapeutic strategies should be developed.

In this study, we discuss the biogenesis and characteristics of exosomes, focusing on mechanisms through which exosomes mediate ferroptosis in the pathogenesis of diseases. In addition, we identify the opportunities and challenges in the state of the art of exosome-based FT to provide insights into exosome-mediated ferroptosis and develop new disease treatment strategies.

## 2. Biogenesis and Characteristics of Exosomes

Although the exact process remains unclear, a three-step multivesicular bodies (MVB)/intraluminal vesicle (ILV) model is generally accepted to understand the biogenesis process of exosomes (Figure 1). To begin with, the first invagination of the plasma membrane packages proteins, lipids, and metabolites associated with the extracellular milieu into cells [25], leading to the formation of early-sorting endosomes (ESEs). In the second step, ESEs communicate with the trans-Golgi network (TGN) and endoplasmic reticulum (ER) to exchange endocytic cargo and luminal constituents [26] and then mature into late-sorting endosomes (LSEs). In the final step, the second invagination of the plasma membrane occurs, leading to the formation of MVBs containing ILVs [27]. MVBs can either fuse with lysosomes or autophagosomes to undergo degradation or with the plasma membrane to release ILVs into the extracellular milieu in the form of exosomes [28]. Exosomes immediately enter the recipient cell by fusing with the membrane, recognizing receptors or endocytosed by target cells [29]. The aforementioned invagination, fusion, and secretion processes lead to the formation of exosomes that facilitate intercellular communication.

Various biophysical and biochemical methods for the isolation of exosomes have been developed [30], including those based on ultracentrifugation, size, immunoaffinity capture, and microfluidics. A previous review detailed the advantages and drawbacks of each technique [31]. The choice of method for exosome isolation depends on the corresponding experimental results and the end use of exosomes. The International Society for Extracellular Vesicles proposed that a combination of methods may outperform individual ones [32]. After isolation, exosomes are characterized by global quantifications, general characterization, and further characterization of single vesicles. Detailed identification standards can be found in MISEV2018 [32].

**Figure 1 cells-12-00311-f001:**
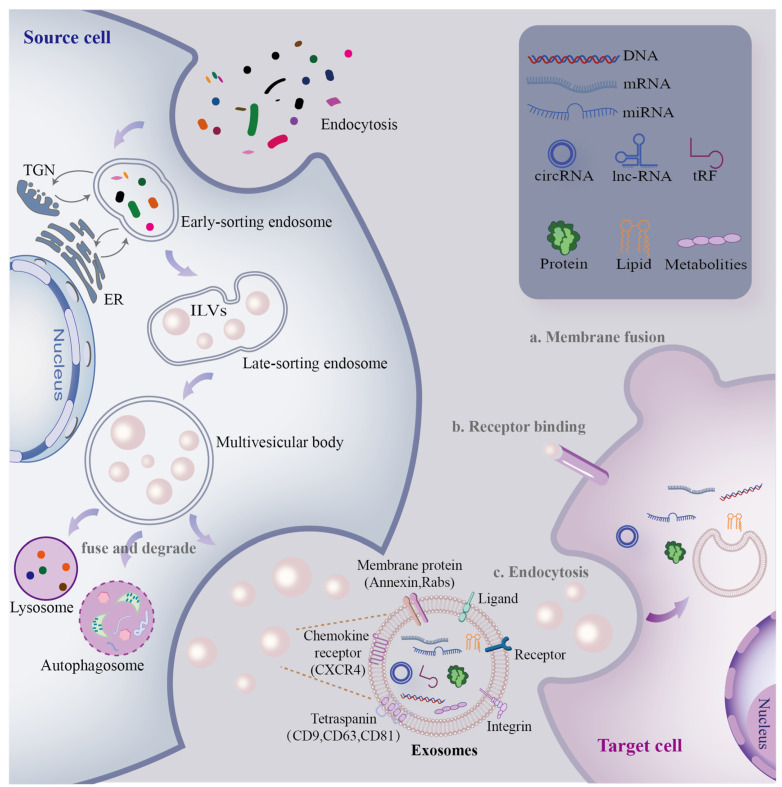
Biogenesis and secretion of exosomes. The secretion of exosomes involves two invaginations of the plasma membrane. Intercellular communication mediated by circulating exosomes occurs through endocytosis, ligand–receptor interaction, and membrane fusion. Components such as proteins, nucleic acids, lipids, and metabolites are delivered by exosomes from the source to target cells [33]. Proteins located in the plasma membrane, including CD63, CD9, CD81, TSG101, and Alix, are commonly used as markers for exosomes [26].

## 3. Role of Exosome-Mediated Ferroptosis in the Pathogenesis of Diseases 

The ferroptosis-related pathogenesis of diseases is always accompanied by an imbalance in intracellular iron homeostasis, dysfunction in lipid oxide metabolism, and by low cellular antioxidative capacity. Although the complex mechanism underlying ferroptosis in diseases is still not completely understood, exosomes regulate proteins related to the ferroptosis pathway by delivering molecular entities (ncRNAs and proteins) or trafficking ferritin-bound iron in a paracrine manner (Figure 2). Thus, researchers must investigate the role of exosomes in ferroptosis-related pathogenic mechanisms.

### 3.1. Regulation of Ferroptosis-Related Pathogenic Mechanisms Is Dependent on Exosomal Cargo

Exosomal RNAs are circulating factors that can be utilized as indicators of diseases because of their unique and diverse types. In various diseases, exosomes exert different effects on ferroptosis by transferring their functional cargo (Table 1).

During the progression of tissue damage and organ dysfunction triggered by ferroptosis, exosomes promote ferroptosis by regulating GSH synthesis and iron metabolism. In long-term high-fat diet (HFD)–induced obesity, exosomes derived from adipose tissue macrophages (ATM-Exos) was noted to cause cardiac injury accompanied by high levels of lipid peroxides. An abundance of miR-140-5p in ATM-Exos inhibited GSH synthesis and promoted ferroptosis in cardiomyocytes by targeting solute carrier family 7 member 11 (SLC7A11) [34]. SLC7A11 regulates glutamine by taking in cysteine and excreting glutamate in a 1:1 ratio [50]. SLC7A11 (a light-chain subunit) and SLC3A2 (a heavy-chain subunit) constitute the cysteine–glutamate antiporter system (system Xc^−^). In the murine model of acute lung injury (ALI), exosomes derived from lipopolysaccharide (LPS)-induced macrophages (LPS-exos) inhibited cell viability and GPX4 and FTH1 expression and upregulated ROS generation in alveolar epithelial cells. TRF-22-8BWS7K092, which was significantly increased in LPS-exos, induced ferroptosis and contributed to the pathogenesis of ALI by activating the Hippo signaling pathway [35]. Thus, the Hippo signaling pathway may be a new therapeutic target for FT-based ALI. The findings of the aforementioned studies have indicated that endogenous exosomes contain molecules that can inhibit GSH synthesis or reduce GPX4 expression. 

Exosomes participate in iron metabolism by regulating divalent metal transporter 1 (DMT1) [51] or glutamate oxaloacetate transaminase 1 (GOT1) [52]. DMT1 facilitates iron uptake by the apical cell membrane and transports iron across the endosomal membrane in almost all cell types that uptake iron through the transferrin (TF) / transferrin receptor protein 1 (TFRC) pathway [53]. GOT1 is the only known iron export protein that transports iron between different types of cells [54]. In a rat model, serum-derived exosomal lncNEAT1 exacerbated sepsis-associated encephalopathy (SAE) by promoting ferroptosis through the miR-9-5p/TFRC and GOT1 axis [36], and NEAT1 is a possible biomarker for the identification of sepsis-induced ferroptosis and SAE. In conclusion, knowledge of the induction of ferroptosis by endogenous exosomes broadens our understanding of disease pathogenesis. Moreover, the inhibition of ferroptosis can prevent and delay tissue damage.

Tumor resistance to chemotherapy might be associated with the ferroptosis-related signaling pathway [55,56,57]. In cancer treatment, the dysregulation of ferroptosis often leads to chemoresistance and treatment failure [58], and exosomal proteins and nc-RNAs derived from the tumor microenvironment exert an antiferroptotic effect on cancer cells to reduce tumor sensitivity to chemotherapeutics and immunotherapy. The aberrant activation of certain abnormal signaling pathways, such as the Wnt/beta-catenin pathway, in cancer cells suppresses ferroptosis by targeting GPX4 [59]. GPX4 and system Xc^−^ are expressed at high levels in colorectal cancer (CRC) and are associated with poor prognosis. Adipose-derived exosomal microsomal triglyceride transfer protein (MTTP) reversed ferroptosis in CRC cells (SW480 and HCT116), accompanied by an increased expression of GPX4 and xCT [37]. Adipose-derived exosomal MTTP inhibited ferroptosis and promoted chemoresistance in CRC through the MTTP/PRAP1/ZEB1 axis. For the treatment of gastric cancer (GC), cisplatin and paclitaxel promoted cancer-associated fibroblasts (CAFs) to secrete exosomal miR-522, leading to decreased ALOX15 and ROS levels and ferroptosis suppression [38]. The knockdown of miR-522 by inhibitors significantly increased cell death (SGC7901, MGC803, and MKN45 cells). The inhibition of miR-522 secretion from CAFs constitutes a novel method for GC treatment. However, miR-522, which served as a tumor-driving factor in that study, is widely expressed in normal tissues. New methods for targeting miR-522 in specific tissues should be developed. Exo-lncFERO derived from GC cells enhanced the stemness of GC stem cells by suppressing ferroptosis [39]. Heterogeneous nuclear ribonucleoprotein A1 (hnRNPA1) was involved in the packing of ncRNAs into exosomes in GC cells. These findings jointly elucidate the chemoresistance of cancer cells and indicate that the blocking of exosomes can aid the sensitization of cells to ferroptosis.

The role of ferroptosis in the pathogenesis of fibrotic tissues, especially liver fibrosis (LF), has been widely studied, and exosomes can interfere with the LF process by regulating the ferroptosis of hepatic stellate cells (HSCs). Increased levels of exosomal miR-222 derived from HBV-infected LO2 cells were observed to promote liver fibrosis by enhancing LX-2 cell activation and inhibiting TFRC-induced ferroptosis of LX-2 [40], a unique molecular mechanism of LF at the cellular level. Consistent with the finding of a previous study [60], the finding of this study demonstrated that miR-222 can serve as a biomarker for HBV-induced hepatic fibrosis, and more clinical studies are needed to confirm the sensitivity and specificity of this liquid biopsy. Interestingly, exosomes from hucMSCs induced the ferroptosis of HSCs by regulating the xCT/GPX4 axis [61]. Thus, regulating the activation and proliferation of HSCs from the perspective of ferroptosis is an effective strategy for LF treatment.

### 3.2. Regulation of Ferroptosis-Related Pathogenic Mechanisms Is Dependent on Trafficking Ferritin-Bound Iron by Exosomes

Ferroptosis-related pathogenesis is dependent on exosomal iron efflux. The relative stability of the labile iron pool (LIP) in cells is maintained by the coordination of iron uptake, utilization, storage, and export. Ferrous iron (Fe^2+^) oxidized by ceruloplasmin (CP) binds to transferrin (TF) to form a Fe^3+^–TF complex, which binds to TFRC and is located in endosomes under endocytosis [62]. In endosomes, the six-transmembrane epithelial antigen of prostate 3 (STEAP3) reduces Fe^3+^ to Fe^2+^, which is favored by increased STEAP3 levels [63]. Excessive iron is stored in ferritin [64], a protein complex composed of ferritin light chain (FTL) and ferritin heavy chain 1 (FTH1), and finally released by FPN [65].

Ferritin maintains tissue and body iron homeostasis through an exosome-dependent pathway. For example, in the murine model of asbestos-induced mesothelial carcinogenesis, ferritin is a major component of ferroptosis-dependent EVs (FedEVs) secreted by macrophages under continual ferroptosis. After EVs are secreted, ferritin is delivered to mesothelial cells and finally contributes to mesothelial carcinogenesis [66]. Ferritin promotes the secretion of CD63^+^ extracellular vesicles (EVs) through the IRE-IRP system. When extracellular vesicles are secreted, ferritin is transferred from iron-loaded cells to CD63^+^ EVs through nuclear receptor coactivator 4 (NCOA4) [67]. NCOA4 mediates the autophagic degradation of ferritin to release iron in LIP [68], and the blockage of NCOA4 reduces the LIP level and inhibits ferroptosis [69]. Exosomes from endothelial cells protected osteoblasts from ferroptosis by inhibiting ferritinophagy [70]; however, the underlying molecular mechanism remains unclear. Ferritin is the key factor that exosome-mediated export inhibits ferroptosis that would otherwise promote ferroptosis. Future studies should explore how exosomes target ferritinophagy.

The discovery of the interaction between prominin-2 and ferritin has key implications for the study of exosomal iron efflux. Under ferroptotic stress, prominin-2 stimulated the formation of MVBs and exosomes and mediated the secretion of ferritin-bound iron from cells into exosomes and thus resisted ferroptosis [71]. On the basis of this mechanism, a study designed a biocompatible hybrid nanoparticle, which was composed of iron oxide nanoparticles, polymers with oxaliplatin attached, and siprominin-2 to inhibit the secretion of exosomes and restore the intracellular iron concentration, thus rescuing ferroptosis resistance and enhancing ferroptosis-based cancer therapy [72]. Nanoplatforms with antiexosomal iron efflux can simultaneously enhance the antitumor immune response. In conclusion, these findings indicate that exosomes regulate ferroptosis by transporting ferritin-bound iron out of the cell; this constitutes an interesting pathway through which cells resist ferroptosis by dynamically exporting ferritin and iron.

## 4. Exosome-Mediated Ferroptosis in Disease Treatment

Endogenous exosomes, as an intercellular communication medium, promote the occurrence and development of diseases mediated by ferroptosis. In addition, exogenous exosomes can affect ferroptosis and thus be used to treat diseases. In this section, we summarize how native and engineered exosomes exert therapeutic effects on tissue regeneration and cancer by targeting ferroptosis.

### 4.1. Therapeutic Effects of Native Exosomes

MSC-derived exosomes (MSC-Exos) have many advantages, including low immunogenicity, high biocompatibility, and stability [73], and they have been widely applied in tissue regeneration [74]. In acute liver injury, myocardial I/R injury, and nervous system diseases, MSC-Exos can regulate ferroptosis to promote tissue repair and regeneration.

Ferroptosis inhibition is a promising strategy for acute liver injury [75]. In a CCl_4_-induced AKI model, MSC-Exos inhibited ferroptosis by rescuing the OTUB1-mediated deubiquitination of SLC7A11, leading to a decrease in the levels of lipids, ROS, and MDA and ultimately alleviating liver damage [76]. Consistent with previous studies [77,78] have indicated that interventions against SLC7A11 or GSH-GPX4 axis inhibited ferroptotic cell death and attenuated tissue injury. In addition to elevated OTUB1 (a deubiquitinase), the exosome-induced recovery of SLC7A11 was accompanied by increased CD44 expression [76]. The hepatic uptake of CD44, which is expressed in MSC-Exos, is responsible for recruiting MSC-Exos to the injured liver. However, whether an interaction occurs between CD44 and OTUB1 remains to be investigated. Moreover, other components in exosomes might play a role in regulating downstream proteins. Compared with unmodified MSC-Exos, exosomes derived from baicalin-pretreated MSCs were observed to exert a more favorable effect on liver damage in the D-GaIN/LPS-induced AKI model by activating the P62-Keap1-NFF2 pathway [79]. Therefore, MSC-Exos can be used to treat ALI, and the therapeutic efficacy of MSC-Exos should be optimized.

Ischemia leads to the accumulation of iron in the myocardium, and iron overload in the myocardium worsens I/R injury [80]. MSC-Exos inhibit the ferroptosis of myocardial cells after acute myocardial infarction (AMI) and promote injury repair. In the infarcted myocardium, exosomal miR-23a-3p of MSCs derived from human umbilical cord blood (HUCB-MSCs) inhibited ferroptosis and attenuated myocardial injury by targeting DMT1 [45].. Interestingly, changes in DMT1 expression had no effect on GPX4 activity, whereas the overexpression or knockdown of DMT1 significantly affected ROS, Fe^2+^, and MDA levels and the level of iron deposition, indicating that HUCB-MSC-derived exosomes inhibited ferroptosis by regulating iron metabolism independent of the GPX4 pathway. Bone marrow mesenchymal stem cell-derived exosomal lncRNA Mir9-3hg exerted cardioprotective effects on I/R-induced cardiac injury by inhibiting cardiomyocyte ferroptosis through modulating Pum2/PRDX6 [46]. Therefore, MSC-Exo can treat AMI by controlling the process of ferroptosis in cardiomyocytes.

The nervous system is closely related to ferroptosis because of its high oxygen consumption and a large store of nonheme iron [81]. The protection of neurons through the inhibition of ferroptosis can serve as a new treatment strategy for neurological diseases. However, because of the limitations of the blood–brain barrier (BBB), the delivery of ferroptosis inhibitors to the brain has been a major challenge in clinical practice. Fortunately, exosomes have the ability to cross the BBB, and native exosomes from MCSs have been considered as an alternative for cell therapy to suppress neuronal cell ferroptosis [82]. The MSC-Exo lncGm36569 inhibited neuronal cell ferroptosis and attenuated neuronal dysfunction through the IncGm36569/miR-5627-5p/FSP1 axis in acute spinal cord injury [44]. FSP1 encoded by apoptosis-inducing factor mitochondrial 2 is another glutathione-independent ferroptosis resistance factor in addition to GPX4 [83,84]. Moreover, MCSs can be genetically modified to overexpress functional molecules, and in this way, exosomes secreted from the parent cell contain the target cargo. Decreased miR-19b-3p expression in intracerebral hemorrhage (ICH) mice was concomitant with neuronal injury [47]. Adipose-derived stem cells (ADSCs) transfected with miR-19b-3p mimic (ADSCs-19bM) derived exosomes attenuated ICH-induced ferroptosis in vivo by directly targeting the iron regulatory protein IRP2, which enhanced the accumulation of iron [85]. IRP2 silencing reduced the level of the iron import protein TfR1 and increased the level of the iron export protein ferroportin (FPN). Additional studies should be conducted to explore whether iron storage regulated by TfR1 or FPN is associated with the endocytosis of exosomes in the mouse model of ICH. In conclusion, exosomes may treat neurological diseases by regulating ferroptosis.

### 4.2. Therapeutic Applications of Engineered Exosomes

With the development of nanotechnology and the physicochemical properties of nanomaterials, various nanoparticles are widely used in research on cancer treatment. The US Food and Drug Administration has approved the use of iron oxide nanoparticles as magnetic resonance imaging (MRI) contrast agents [86] and drug carriers. Gadolinium-based contrast agents are mainly used in MRI, but they harm the liver and kidney [87,88]. Iron oxide nanoparticles, especially superparamagnetic iron oxide nanoparticles, can be used as contrast agents in MRI [89]. Compared with gadolinium-based contrast agents, iron oxide nanoparticle-based contrast agents have superior biocompatibility and safety and achieve higher spatial resolution, thus improving diagnostic accuracy [90]. Moreover, iron oxide nanoparticles can be used as MRI contrast agents because of their low toxicity and amenability to surface modification [91]. Therefore, iron oxide nanoparticles are more favorable for use in treatment than as an imaging contrast agent.

Iron-based nanomaterials induce ferroptosis in cancer cells by increasing ROS production through the Fenton reaction and by depleting glutathione. Shen et al. designed magnetic nanoparticles called FeGd-HN@Pt@LF/RGD2 [92], which could be internalized into cancer cells. Fe^2+^ and Fe^3+^ released by nanoparticles in situ produced ROS and accelerated the Fenton reaction in the tumor site, leading to the substantial inhibition of tumor growth. Another study developed copper–iron oxide spinel nanoparticles that reduced the GSH level and generated ROS [93]. It reveals that iron-based nanomaterials have the potential to enhance the efficacy of ferroptosis-based cancer therapy.

Among the many nanomaterials that have been developed, exosomes are promising owing to their chemical stability and biocompatibility, and they can act as endogenous functional biomolecules with the advantages of nontoxicity, low immunogenicity, and the ability to cross the BBB. Physicochemical and genetic engineering methods are usually used to incorporate cargo in exosomes [94]. As mentioned above, the BBB limits the use of ferroptosis inhibitors in patients with neurological diseases. However, the safe and effective delivery of drugs to the brain by exosomes is a promising new approach for the treatment of neurological diseases [95]. Besides, ref. [96] due to the rapid development of nanotechnology, various nanocarrier-mediated drug delivery systems are emerging as effective methods for the treatment of glioma. These nanocarriers can mediate the delivery of drugs across the BBB to target tumor sites. In conclusion, exosomes and nanomaterial-mediated drug delivery systems can lead to better diagnosis and treatment than traditional cancer treatments (e.g., radiotherapy and chemotherapy).

A study designed a three-step approach-based exosome to induce ferroptosis, involving DHODH and GPX4 ferroptosis defense mechanisms and Fe_3_O_4_ NP-mediated Fe^2+^ release. First, genetic engineering methods were used to add the Angiopep-2 (ANG) peptide to the fusion gene of Lamp2b (a signal peptide present on the surface of exosomes). The obtained ANG peptide–modified engineered exosomes had stronger BBB penetration and brain-targeting abilities [97]. Subsequently, small interfering RNA (siRNA) of GPX4 (siGPX4) was loaded into exosomes through electroporation. This study designed a complex with two components: ANG peptide–modified engineered exosomes and a Fe_3_O_4_-core mesoporous-silica shell conjugated with a CD63 antibody. Finally, the complex was incubated with brequinar, an inhibitor of DHODH. This study developed a new platform for glioblastoma therapy. A recent study developed *FAP* gene–engineered tumor cell-derived exosome-like nanovesicles as a tumor vaccine that can target the tumor parenchyma, enhance the immune response, and promote tumor ferroptosis. Therefore, the modification of ferroptosis key genes expressed in exosomes effectively induced ferroptosis in tumor cells.

One other method centers on the production of drug-loaded exosomes. Exosomes have been applied in many cancer types for drug delivery. The classic ferroptosis inducer erastin (Er) is poorly soluble in water and nephrotoxic; therefore, a drug delivery system with high efficiency and few side effects should be developed. Du et al. chose an exosome-based platform [98]. They loaded exosomes with Er and the photosensitizer Rose Bengal (RB) to treat hepatocellular carcinoma. RB and Er cooperated to generate more ROS locally. Moreover, exosomes expressing CD47 escaped the phagocytosis of the mononuclear phagocyte system, thus increasing the distribution of exosomes in tumor tissues. To increase the selectivity of exosomes in TNBC cells, Yu et al. modified TNBC cells with FA [99], a specific ligand for the FA receptor that is overexpressed in TNBC cells. Erastin@FA-exo promoted ferroptosis by depleting glutathione and overgenerating ROS. Unlike other ferroptosis inducers that usually mediate a single pathway, a low Er concentration in exosomes provides greater efficacy by mediating multiple pathways. 

## 5. Discussion

A unified standard for the identification of ferroptosis should be established. Ferroptosis is defined as a process of cell death suppressed by both iron depletion and lipophilic antioxidants [1]. However, there are some iron-dependent lethal mechanisms that differ from ferroptosis, and some oxidative stress mechanisms do not depend on iron [100]. Thus, the suppression of cell death by a single inhibitor of ferroptosis does not constitute solid evidence for the involvement of ferroptosis in the process of interest. A standard for the identification of ferroptosis and precise biomarkers for in vivo ferroptosis, such as cleaved caspase-3 for apoptosis, should be developed.

Considerable time and research would be required before exosomes can be used as cell-free therapy in clinical practice. First, the efficient isolation of exosomes, the reduction in high heterogeneity caused by donor cells, and the prevention of degradation during long-term storage are crucial. Although ultracentrifugation is the gold standard for the isolation of exosomes currently, it has low isolation efficiency and leads to low-quality exosomes, and centrifugal force can destroy the structure and function of exosomes. Large-scale, randomized, placebo-controlled clinical trials should be performed to evaluate the safety and long-term effects of exosomes to achieve positive therapeutic effects. Recent major advances in exosome research have been based on the growing recognition that EVs, including exosomes, have many different particle isoforms and that each of these isoforms may play specific roles in cellular communication [101]. However, despite the growing interest in this subfield, our understanding of the cellular and molecular mechanisms that control extracellular vesicle biogenesis, release, uptake, and function remains limited.

Although exosomes have been demonstrated to play an important role in the progress and treatment of diseases, several barriers require extra effort. The poor knowledge of molecular mechanisms supporting exosome-mediated ferroptosis limits the rational exploitation of exosomes in FT. It is important to confirm the exosome imports cystine and export glutamine because a high expression level of SLC7A11 protects cells from ferroptosis by promoting the uptake of cysteine to synthesize reduced GSH. Although studies have demonstrated that exosomes regulate SLC7A11, the exact mechanism underlying this regulation remains to be elucidated. We explored whether exosomes, in addition to regulating SLC7A11, regulate other molecules to affect GSH synthesis, thus intervening in the ferroptosis process. Moreover, further studies are required to examine how exosomes export iron from iron-overloaded cells. A common molecular mechanism exists between exosomes and autophagy. Thus, iron efflux under ferroptotic stress may be associated with autophagy. In addition to the most commonly studied NCOA4-mediated ferritinophagy [102], more attention should be focused on other types of autophagy, such as RAB7A-mediated lipophagy, STAT3-induced lysosomal membrane permeabilization, and HSP90-associated chaperone-mediated autophagy, which are involved in ferroptosis [103]. Ferroptosis acts as a double-edged sword, especially in tumors; studies should differentiate between ferroptosis that inhibits tumor growth, and ferroptosis that drives cancer progression. In conclusion, a better understanding of the molecular events of ferroptosis can help develop appropriate methods to intervene in ferroptosis by exosomes. 

The combination of exosome-based FT and other therapies is the future development trend. Exosome-based FT combined with immune checkpoint blockade and radiation therapy can rescue antitumor immunity and diminish exosomal suppression. For example, the assembly of an exosome inhibitor (GW4869) and a ferroptosis inducer (Fe^3+^) by the use of amphiphilic hyaluronic acid enhanced antitumor immune activation [104,105]. Exosomes are essential for desensitizing cancer cells to ferroptosis; blocking endogenous exosomes is a promising strategy to induce ferroptosis and improve the efficacy of chemotherapy for future cancer treatment. It is promising to be generalized to develop novel exosomal immune therapeutics against malignant metastatic tumors by targeting ferroptosis.

## 6. Conclusions

Endogenous and exogenous exosomes play a pathogenic or therapeutic role by regulating ferroptosis. The functional role of exosomes in ferroptosis is summarized in three ways (Figure 3). The regulation of ferroptosis-related pathogenic mechanisms is dependent on exosomal proteins and ncRNAs. Under ferroptotic stress, exosomes secreted from tumor cells can export iron to maintain intracellular iron concentration and thus resist ferroptosis. When used for disease treatment, native and bioengineered exosomes both exert positive effects by targeting ferroptosis. Therefore, future studies should focus on developing exosome-based FT for treatment.

## Figures and Tables

**Figure 2 cells-12-00311-f002:**
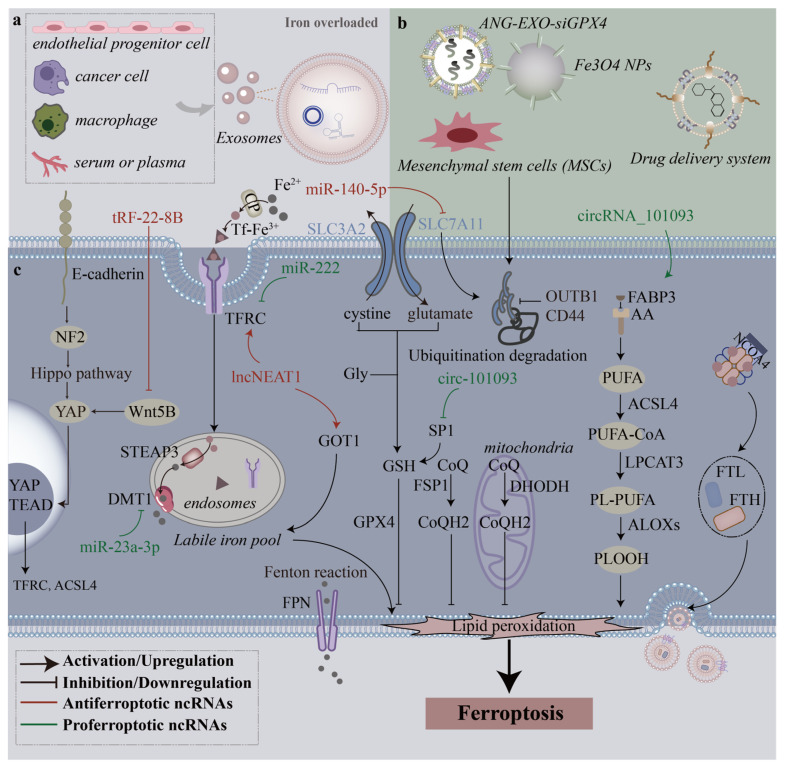
Mechanisms through which different sources of exosomes regulate ferroptosis. (**a**), In the iron-overloaded milieu, many types of cells and even serum or plasma release endogenous exosomes that affect ferroptosis. (**b**), Various exosomes derived from mesenchymal stem cells (MSCs) and engineering exosomes can be used to treat diseases by modulating ferroptosis. (**c**), Exosome-derived ncRNAs regulate multiple intracellular ferroptosis pathways to promote or inhibit ferroptosis.

**Figure 3 cells-12-00311-f003:**
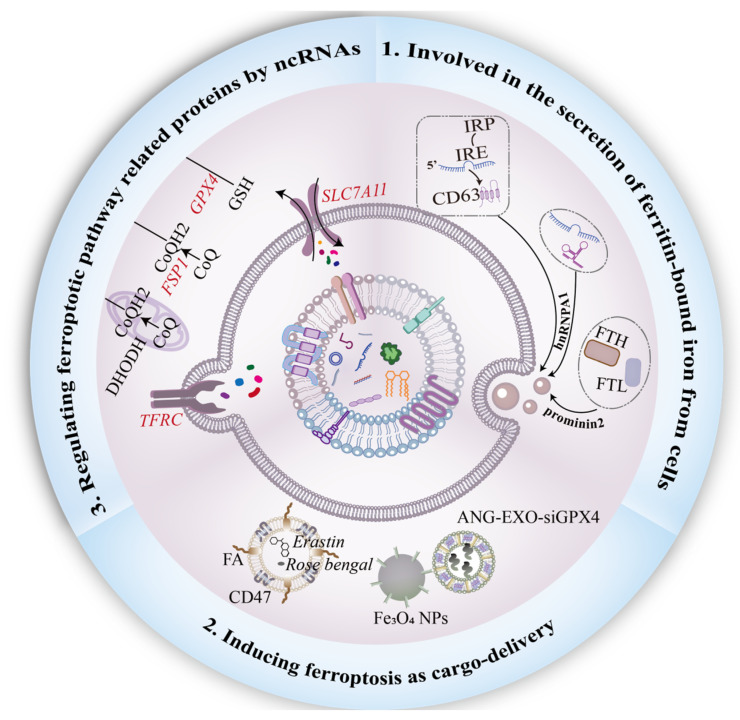
Functional role of exosomes in ferroptosis. First, the interaction between prominin-2 and ferritin promotes the secretion of ferritin-bound iron from cells into exosomes. Heterogeneous nuclear ribonucleoprotein A1 (hnRNPA1) is involved in the process of ncRNAs packed into exosomes, and the IRE-IRP system plays an indispensable role in the secretion of CD63^+^ EVs. Moreover, Engineering exosomes are usually designed for ferroptosis-inducer delivery. To sensitize tumor cells to ferroptosis, genetic engineering or chemical modification on the surface of exosomes should be employed. Finally, to combat ferroptosis, exosomal ncRNAs regulate ferroptotic-related proteins, including SLC7A11, FSP1, and TFRC, in a paracrine manner.

**Table 1 cells-12-00311-t001:** Exosomal cargo functioning in diseases by mediating ferroptosis.

Sources	Exosomal Cargo	Diseases	Ferroptotic Cells	Target	Reference	Findings on Outcomes
ATMs	miR-140-5p	Obesity-induced cardiac injury	H9c2	SLC7A11	[34]	Inhibited GSH synthesis by targeting SLC7A11 and induced ferroptosis and cardiac injury
Alveolar macrophages	tRF-22-8B	ALI	MLE-12	Hippo pathway	[35]	Inhibited GPX4 and FTH1, enhanced oxidative stress, induced ferroptosis, and contributed to ALI pathogenesis
Serum	lncNEAT1	SAE	bEnd.3	TFRC and GOT1	[36]	Exacerbated SAE by promoting ferroptosis through regulating the miR-9-5p/TFRC and GOT1 axis
Adipocytes	MTTP	CRC	SW480, HCT116	/	[37]	Reduced susceptibility to ferroptosis in CRCs, thus promoting chemoresistance to oxaliplatin
CAFs	miR-522	GC	SGC7901, MGC803, MKN45	ALOX15	[38]	Inhibited ferroptosis and lipid and ROS accumulation in cancer cells and ultimately reduced chemosensitivity
Gastric cancer cells	lncFERO	GC	SGC7901, MKN45	/	[39]	Enhanced stemness and acquired chemoresistance by suppressing ferroptosis and targeting exo-lncFERO/hnRNPA1/SCD1 axis
HBVs-infected hepatocytes	miR-222	Liver fibrosis	LX2	TFRC	[40]	Promoted liver fibrosis by inhibiting TFRC and TFRC-induced ferroptosis
Plasma	circ-101093	LUAD	H1650, PC9, H1975, H358, A549, H1299	FABP3-AA	[41]	Reduced global AA and desensitized cells to ferroptosis
Tumor lung tissues	miR-4443	NSCLC	A549	FSP1	[42]	Inhibited FSP1-mediated ferroptosis induced by cisplatin treatment and promoted the chemoresistance of NSCLC
Hypoxic lung cancer cells	ANGPTL4	NSCLC	A549, H1299	/	[43]	Inhibited ferroptosis of normoxic cells and reduced their radiosensitivity
MSCs	lncGm36569	ASCI	HT-22	FSP1	[44]	Inhibited ferroptosis through the miR-5627-5p/FSP1 axis and attenuated neuronal dysfunction
HUCB-MSCs	miR-23a-3p	AMI	Primary myocardial cell	DMT1	[45]	Inhibited ferroptosis and attenuated myocardial injury
BMSCs	lncMir9-3hg	Cardiac I/R injury	HL-1	/	[46]	Inhibited cardiomyocyte ferroptosis through the Pum2/PRDX6 axis and ameliorated cardiac function
ADSCs	miR-19b-3p	ICH	Primary Neurons	IRP2/TfR1/FPN	[47]	Attenuated hemin-induced cell injury and ferroptosis
Cardiac fibroblasts	miR-23a-3p	Atrial fibrillation	H9c2	SLC7A11	[48]	Exacerbated ferroptosis in h9c2 cells by targeting SLC7A11
EPCs-Exos	miR-30e-5p	Vascular endothelial injury	HUVECs	SP1 and AMPK pathway	[49]	Activated the AMPK pathway by miR-30e-5p targeting SP1 and inhibited erastin-induced HUVEC ferroptosis

Abbreviations: CAFs, cancer-associated fibroblasts; GC, gastric cancer; LUAD, lung adenocarcinoma; NSCLC, nonsmall cell lung carcinoma; ANGPTL4, angiopoietin-like 4; MTTP, microsomal triglyceride transfer protein; CRC, colorectal cancer; ASCI, acute spinal cord injury; I/R, ischemia reperfusion; HUCB-MSCs, MSCs derived from human umbilical cord blood; AMI, acute myocardial infarction; ATMs, adipose tissue macrophages; EPC-Exos, human umbilical vein blood endothelial progenitor cells-exosomes; HUVEC, human umbilical vein endothelial cells; SAE, sepsis-associated encephalopathy; ALI, acute lung injury; ADSCs, adipose-derived stem cells; ICH, intracerebral hemorrhage; AA, arachidonic acid.

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
