# Peer review of "Implications of Crosstalk between Exosome-Mediated Ferroptosis and Diseases for Pathogenesis and Treatment"

_cells, 2023, doi:10.3390/cells12020311_

Round 1
Reviewer 1 Report
Well-rounded review of the material. Writing style was a pleasure to read. Very informative, great references and figures.
Author Response
Jan 4, 2023
Cells
Manuscript ID: cells-2092147
Title: Implications of crosstalk between exosome-mediated ferroptosis and diseases for pathogenesis and treatment
Authors: Zixuan Zhou1,2#, Benshuai You1,2#, Cheng Ji1,2, Leilei Zhang1,2, Feng Wu1,2, Hui Qian1,2*
Dear Editor,
Thank you for your e-mail dated Dec 28, 2022 regarding the review of our manuscript. We appreciate your assessments of the manuscript and have found that the comments and suggestions are helpful in preparation of the revised manuscript. We revised the manuscript as suggested by reviewers. In the revised manuscript, the references format has been revised. The following are our point-by-point responses, in order of the comments about the manuscript:
Reviewer1
Q1: Well-rounded review of the material. Writing style was a pleasure to read. Very informative, great references and figures.
Responses: We gratefully thanks for the precious time the reviewer spent making positive comments.
Thank you for your consideration. We look forward to hearing from you and to publication of this manuscript.
Sincerely,
Hui Qian, Prof.
School of Medicine, Jiangsu University,
301 Xuefu Road, 212013, Zhenjiang, Jiangsu, P.R. China.
E-mail: lstmmmlst@163.com;
Reviewer 2 Report
Ferroptosis is a type of iron-dependent cell death caused by ferrous iron overload, reactive
oxygen species generation through the Fenton reaction, and lipid peroxidation, leading to antioxi-
dative system dysfunction and ultimately cell membrane damage. The functional role of ferroptosis
in human physiology and pathology is considered a cause or consequence of diseases. Circulating
exosomes mediate intercellular communication and organ crosstalk. They not only transport func-
tional proteins and nucleic acids derived from parental cells but also serve as vehicles for the tar-
geted delivery of exogenous cargo. Exosomes regulate ferroptosis by delivering the biological ma-
terial to the recipient cell, affecting ferroptosis-related proteins, or transporting ferritin-bound iron out of the cell. This review discusses pathogenesis mediated by endogenous exosomes and the therapeutic potential of exogenous exosomes for ferroptosis related diseases. In addition, this review explores the role of exosome-mediated ferroptosis in ferroptosis-related diseases with an emphasis
on strategies for engineering exosomes for ferroptosis therapy.
1. Figs are low
2. Citation is low
3. The discusion is not good
4. The novelty is low
5. The goal is not considered well
Author Response
Jan 4, 2023
Cells
Manuscript ID: cells-2092147
Title: Implications of crosstalk between exosome-mediated ferroptosis and diseases for pathogenesis and treatment
Authors: Zixuan Zhou1,2#, Benshuai You1,2#, Cheng Ji1,2, Leilei Zhang1,2, Feng Wu1,2, Hui Qian1,2*
Dear Editor,
Thank you for your e-mail dated Dec 28, 2022 regarding the review of our manuscript. We appreciate your assessments of the manuscript and have found that the comments and suggestions are helpful in preparation of the revised manuscript. We revised the manuscript as suggested by reviewers. In the revised manuscript, the references format has been revised. The following are our point-by-point responses, in order of the comments about the manuscript:
Reviewer2
Q1: Figs are low. Citation is low. The discussion is not good. The novelty is low. The goal is not considered well.
Responses: We are very grateful to your comments for the manuscript. Our review is a novel piece of work dealing with a new concept of ferroptosis and the involvement of exosomes. This review clearly explains most of the salient points related to this study starting from a brief description about ferroptosis and exosomes and clarifying the role of exosomes in the process of ferroptosis in various disease conditions. It covers most of recent progress made in this emerging area of research.
Thank you for your consideration. We look forward to hearing from you and to publication of this manuscript.
Sincerely,
Hui Qian, Prof.
School of Medicine, Jiangsu University,
301 Xuefu Road, 212013, Zhenjiang, Jiangsu, P.R. China.
E-mail: lstmmmlst@163.com;
Reviewer 3 Report
The review covers a comprehensive review about recurrent prostate cancer that initially responds to androgen therapy to develop castrate resistant disease (CRPC) and progress to the lethal metastatic CRPC (mCRPC). These cancer cells switch their metabolism from glycolysis to oxidative phosphorylation (OXPHOS) that generates excess reactive oxygen species (ROS) that induce mitochondrial dysfunction, autophagy, cancer invasion, and metastasis. Recent progresses in mass spectrometry and hyperpolarized magnetic resonance imaging have now enabled monitoring in vitro and in vivo metabolic changes vivo in tumor tissues. The power of these technologies can now be harnessed in the clinic to personalize metabolism targeted therapies to treat therapy refractory CRPCs and mCRPCs.
Author Response
Jan 4, 2023
Cells
Manuscript ID: cells-2092147
Title: Implications of crosstalk between exosome-mediated ferroptosis and diseases for pathogenesis and treatment
Authors: Zixuan Zhou1,2#, Benshuai You1,2#, Cheng Ji1,2, Leilei Zhang1,2, Feng Wu1,2, Hui Qian1,2*
Dear Editor,
Thank you for your e-mail dated Dec 28, 2022 regarding the review of our manuscript. We appreciate your assessments of the manuscript and have found that the comments and suggestions are helpful in preparation of the revised manuscript. We revised the manuscript as suggested by reviewers. In the revised manuscript, the references format has been revised. The following are our point-by-point responses, in order of the comments about the manuscript:
Reviewer 3
Q1: The review covers a comprehensive review about recurrent prostate cancer that initially responds to androgen therapy to develop castrate resistant disease (CRPC) and progress to the lethal metastatic CRPC (mCRPC). These cancer cells switch their metabolism from glycolysis to oxidative phosphorylation (OXPHOS) that generates excess reactive oxygen species (ROS) that induce mitochondrial dysfunction, autophagy, cancer invasion, and metastasis. Recent progresses in mass spectrometry and hyperpolarized magnetic resonance imaging have now enabled monitoring in vitro and in vivo metabolic changes vivo in tumor tissues. The power of these technologies can now be harnessed in the clinic to personalize metabolism targeted therapies to treat therapy refractory CRPCs and mCRPCs.
Responses: We gratefully thanks for the precious time the reviewer spent making comments. After reading the referee reports, we think they are not relevant to our manuscript.
Thank you for your consideration. We look forward to hearing from you and to publication of this manuscript.
Sincerely,
Hui Qian, Prof.
School of Medicine, Jiangsu University,
301 Xuefu Road, 212013, Zhenjiang, Jiangsu, P.R. China.
E-mail: lstmmmlst@163.com;
Reviewer 4 Report
In this review article Zhou et al have discussed the pathogenesis mediated by endogenous exosome. Authors have also discussed how exogenous exosomes can be potentially used therapeutically for ferroptosis –related diseases. It is very well written review that covers most of recent progress made in this emerging area of research. I will recommend accepting article for publication in ‘Cells’.
Author Response
Jan 4, 2023
Cells
Manuscript ID: cells-2092147
Title: Implications of crosstalk between exosome-mediated ferroptosis and diseases for pathogenesis and treatment
Authors: Zixuan Zhou1,2#, Benshuai You1,2#, Cheng Ji1,2, Leilei Zhang1,2, Feng Wu1,2, Hui Qian1,2*
Dear Editor,
Thank you for your e-mail dated Dec 28, 2022 regarding the review of our manuscript. We appreciate your assessments of the manuscript and have found that the comments and suggestions are helpful in preparation of the revised manuscript. We revised the manuscript as suggested by reviewers. In the revised manuscript, the references format has been revised. The following are our point-by-point responses, in order of the comments about the manuscript:
Reviewer 4
Q1: In this review article Zhou et al have discussed the pathogenesis mediated by endogenous exosome. Authors have also discussed how exogenous exosomes can be potentially used therapeutically for ferroptosis –related diseases. It is very well written review that covers most of recent progress made in this emerging area of research. I will recommend accepting article for publication in ‘Cells’.
Responses: Thanks for your positive comments.
Thank you for your consideration. We look forward to hearing from you and to publication of this manuscript.
Sincerely,
Hui Qian, Prof.
School of Medicine, Jiangsu University,
301 Xuefu Road, 212013, Zhenjiang, Jiangsu, P.R. China.
E-mail: lstmmmlst@163.com;
Reviewer 5 Report
This article by the Zhou et al., is a novel piece of work dealing with a new concept of ferroptosis and the involvement of exosomes. This review clearly explains most of the salient points related to this study starting from a brief description about ferroptosis and exosomes and clarifying the role of exosomes in the process of ferroptosis in various disease conditions.
From my side I suggest this article is fit for publication.
Author Response
Jan 4, 2023
Cells
Manuscript ID: cells-2092147
Title: Implications of crosstalk between exosome-mediated ferroptosis and diseases for pathogenesis and treatment
Authors: Zixuan Zhou1,2#, Benshuai You1,2#, Cheng Ji1,2, Leilei Zhang1,2, Feng Wu1,2, Hui Qian1,2*
Dear Editor,
Thank you for your e-mail dated Dec 28, 2022 regarding the review of our manuscript. We appreciate your assessments of the manuscript and have found that the comments and suggestions are helpful in preparation of the revised manuscript. We revised the manuscript as suggested by reviewers. In the revised manuscript, the references format has been revised. The following are our point-by-point responses, in order of the comments about the manuscript:
Reviewer 5
Q1: This article by the Zhou et al., is a novel piece of work dealing with a new concept of ferroptosis and the involvement of exosomes. This review clearly explains most of the salient points related to this study starting from a brief description about ferroptosis and exosomes and clarifying the role of exosomes in the process of ferroptosis in various disease conditions. From my side I suggest this article is fit for publication.
Responses: Thank you so much for your positive comments.
Thank you for your consideration. We look forward to hearing from you and to publication of this manuscript.
Sincerely,
Hui Qian, Prof.
School of Medicine, Jiangsu University,
301 Xuefu Road, 212013, Zhenjiang, Jiangsu, P.R. China.
E-mail: lstmmmlst@163.com;
Round 2
Reviewer 2 Report
The text is revised well